# Designing Statistical Models for Holstein Rearing Heifers’ Weight Estimation from Birth to 15 Months Old Using Body Measurements

**DOI:** 10.3390/ani11071846

**Published:** 2021-06-22

**Authors:** Luca Turini, Giuseppe Conte, Francesca Bonelli, Alessio Madrigali, Brenno Marani, Micaela Sgorbini, Marcello Mele

**Affiliations:** 1Department of Veterinary Sciences, University of Pisa, 56124 Pisa, Italy; francesca.bonelli@unipi.it (F.B.); alessio.madrigali@phd.unipi.it (A.M.); brenno.vaevictis@gmail.com (B.M.); micaela.sgorbini@unipi.it (M.S.); 2Centro di Ricerche Agro-Ambientali “E. Avanzi”, University of Pisa, 56122 Pisa, Italy; giuseppe.conte@unipi.it (G.C.); marcello.mele@unipi.it (M.M.); 3Istituto Zooprofilattico Sperimentale del Lazio e della Toscana ‘M. Aleandri’, 00178 Rome, Italy; 4Department of Agriculture, Food and Environment, University of Pisa, 56124 Pisa, Italy

**Keywords:** heifer, estimated body weight, body measure, wither height, shin circumference, heart girth circumference, body length, hip width, body condition score

## Abstract

**Simple Summary:**

The growth monitoring process represents an important part of rearing heifers. The use of a scale is not feasible in some breeding conditions; it may be interesting to investigate the possibility of evaluating body weight (BW) with body measurements. The aim of this study was to estimate heifers’ weight based on their body dimension characteristics. A total of 25 Holstein rearing heifers were monitored after birth, weekly until 2 months of life and monthly until 15 months of age. Animals were weighed, and their wither height (WH), shin circumference (SC), heart girth circumference (HG), body length (BL), hip width (HW) and body condition score (BCS) were measured using tape measure. Equations were built with a stepwise regression to estimate the BW at each time using body measures for the study group. Equations were able to estimate the BW of heifers under a 0.800 kg as an average weight gain target using different variables, representing an alternative method of BW evaluation without a scale. Three variables or fewer were needed for BW estimation at crucial growing times, making these models feasible for use in the field. Different growing rate target may be studied in order to evaluate possible modifications to our equations.

**Abstract:**

Body measurements could be used to estimate body weight (BW) with no need for a scale. The aim was to estimate heifers weight based on their body dimension characteristics. Twenty-five Holstein heifers represent the study group (SG); another 13 animals were evaluated as a validation group (VG). All the heifers were weighed (BW) and their wither height (WH), shin circumference (SC), heart girth circumference (HG), body length (BL), hip width (HW) and body condition score (BCS) were measured immediately after birth, and then weekly until 2 months and monthly until 15 months old. Equations were built with a stepwise regression in order to estimate the BW at each time using body measures for the SG. A linear regression was applied to evaluate the relationship between the estimated BW and the real BW. Equations found were to be statistically significant (r^2^ = 0.688 to 0.894; *p* < 0.0001). Three variables or fewer were needed for BW estimation a total of 11/23 times. Regression analysis indicated that the use of HG was promising in all the equations built for BW estimation. These models were feasible in the field; further studies will evaluate possible modifications to our equations based on different growing rate targets.

## 1. Introduction

Rearing heifers to replace unhealthy or unproductive cows is important for maintaining a productive dairy farm. The cost of rearing heifers is high, representing about 20% of dairy farm expenses, making it the highest variable cost after feed [1]. However, in many dairy herds, the health of adult milking and dry cows is often prioritized over the health and management of youngstock, which are observed less frequently [2]. In order to achieve optimal performance, heifers need to remain healthy, meet target growth rates and be well-grown before first calving [3]. Thus, poor heifer management can lead to economic loss and decreases in animals’ welfare.

Growth rate probably represents the most important parameter for monitoring heifers’ performance throughout the rearing process from birth to calving [4]. Several methods are used to evaluate or estimate body weight (BW) in Holstein heifers; the most accurate method was to weigh animals individually on a scale. However, a dairy farm might not have a scale for weighing heifers on a routine basis, and most dairy producers consider weighing youngstock with a scale to be time-consuming and costly to implement [5]. As a result, rapid and indirect methods of estimating BW have been developed. Body measurements (BM) have been used to predict the growth rate [6], the body condition, and the conformation [7] in beef cattle, monitor the growth of the female, estimate contemporary weight [5], and determine the nutritional requirements of dairy cattle [8]. The most common and cheapest indirect method for BW estimation was the heart girth tape developed by Heinrichs and Hargrove [9]. The authors estimated the BW of 5723 heifers, starting from the heart girth circumference (HG), which was evaluated by placing a measuring tape around the circumference of the animal, just behind the withers. Later, other BMs were observed to successfully estimate BW, such as body length (BL), hip width (HW), and wither height (WH) [10]. Papers evaluating the relationships between BM and BW came from very old studies, and it could be possible that management, feeding and genetic have changed a lot since [5,9,10]. Thus, these equations may not be suitable for modern breeding practices, making the present investigation very relevant. Moreover, these studies are especially focused on lactating dairy cows [11,12,13], without information on rearing animals or a dedicated scale. Due to the importance of monitoring heifers’ growth rate, investigating the relationship between BW and BM in Holstein-rearing heifers could be interesting. The aim of the present study was to estimate dairy heifers’ weight based on their body dimension characteristics, from birth until 15 months of life.

## 2. Materials and Methods

### 2.1. Animals

Our prospective observational study was approved by the Institutional Animal Care and Use Committee of the University of Pisa (OPBA, Pisa, prot. n. 0023045/2018). The owner’s written consent was obtained for an evaluation of the measurements of animals included in the study.

A total of 45 Holstein rearing heifers were recruited from the C.I.R.A.A. “E. Avanzi”, the dairy farm of the University of Pisa. To be included in the study, heifers had to be born from cows with a physiological gestational length (>260 days), and transferred to the maternity unit at least 1 week before calving [14], before ungoing a normal parturition without any manual, pharmacological or surgical assistance. Moreover, heifers had to be vital at birth and healthy based on their history and a physical examination at each measuring time. All the Holstein heifers included in the present study underwent the same management conditions.

### 2.2. Housing

Within 30 min after calving, calves were removed from the dam and housed in a single straw-bedded pen (2.5 m × 2 m) until 8 weeks of life, in accordance with European Legislation (2008/119/CE). Between 9 weeks and 6 months of life, rearing heifers were housed in a collective straw-bedded pen (4.5 m × 3.4 m) in groups composed of three animals. Rearing heifers between 6 and 15 months old were housed in a collective straw-bedded pen (16 m × 10 m) in groups of 20 animals until the positive pregnancy check, then they were moved to a dedicated area of the farm until calving.

### 2.3. Diets

A total of 2 L of good-quality colostrum (≥50 g/L of Ig), evaluated with an optical Brix refractometer (Atago brix N1, Tokyo, Japan), milked from their own dam or from the colostrum bank, was administered as soon as the calf could drink (30 min–2 h). A further 2 L was administered within the next 4–8 h to achieve a successful transfer of passive immunity [15]. All the calves received a total of 2 L colostrum, twice a day, until the third day of life. Then, they received 3 L of whole milk at 39 °C, twice a day until 60 days of life. All the feeding procedures were conducted by an expert operator using a nipple bucket. From the third day of life, fresh and clean water was provided to each calf ad libitum. Free-choice hay was administered after the first week of life. From weaning to 15 months old, calves were fed a diet based on grass hay, alfalfa hay and flacked cereal grain mix, offered as a total mixed ration. The diet was formulated according to the nutritional requirements, calculated using the Cornell Net Carbohydrate and Protein System (CNCPS) [16], considering 0.800 kg as an average weight gain target.

### 2.4. Body Measurements

Rearing heifers were weighed with a scale (ID 3000, Tru-Test Limited, San Diego, CA, USA) at birth, then weekly until 2 months of life and monthly until 15 months of age, for a total of 23 evaluations (T0: at birth, T1: 7 days old, T2: 14 days old, T3: 21 days old, T4: 28 days old, T5: 35 days old, T6: 42 days old, T7: 49 days old, T8: 56 days old, T9: 2 months old, T10: 3 months old, T11: 4 months old, T12: 5 months old, T13: 6 months old, T14: 7 months old, T15: 8 months old, T16: 9 months old, T17: 10 months old, T18: 11 months old, T19: 12 months old, T20: 13 months old, T21: 14 months old, T22: 15 months old). Body measurements took place at the same time as the weighting and were always made by the same operator (AM). In brief, from birth to 6 months old, the animals were manually conducted to the scale in a dedicated area and manually restricted for body measurements. After reaching 6 months of life, rearing heifers were sent through a chute and weigh scale system, and one individual (LT) recorded the actual BW of the animal. The electronic scales were calibrated annually as part of the standard operating procedures of the dairy farm. Subsequently, the animal was released from the weigh scale and put in a locking head gate. The HG was measured as the minimal circumference around the body immediately behind the front shoulder; SC was measured as the smallest circumference of the tibia of the foreleg; WH was the distance from the floor beneath the calf to the top of the withers directly above the center of the shoulder; BL was the distance from the point of the shoulders to the ischium; HW was the widest point at the center of the stifle [10]. Finally, BCS was evaluated, as originally reported by Edmondson and colleagues [17]. All the measurements were made using a plastic-coated fiber tape, which was commercially available in metric graduations (Animeter, Kerbl, Germany).

### 2.5. Statistical Analysis

All statistical evaluations were performed using R software [18]. The study focused on the identification of quantitatively complex relationships between variables: body measurements (SC, BCS, HG, WH, BL and HW) and BW (dependent variable). The research was carried out based on stepwise multiple regression modeling, which allows one to search and describe quantitatively complex relationships. The construction of a multiple linear regression (MLR) model helps to investigate the impact of several independent variables (X1, X2, …, Xk) for one dependent variable (Y). Backward stepwise regression is an extension of linear regression models based on Pearson’s correlation coefficient. Multiple regression model takes the form:Y = β_0_ + β_1_X_1_ + β_2_X_2_ + … + β_k_X_k_ + ε
where Y is the dependent variable; X_1_, X_2_, …, X_k_ are independent variables; β_1_, β_2_, …, β_k_ are parameters; ε is random component (the rest of the model).

The essential value of the MLR models is based on the scientific information contained in the obtained equation. The applied stepwise regression was designed to leave a minimum set of independent variables in the regression model, while maximizing the adjusted determination coefficient and minimizing the mean squared deviation from the regression model. This method involves, as a first step, the construction of a model that contains all potential dependent variables, and then gradually eliminates them in order to maintain the model with the highest determination coefficient, while maintaining the significance of the parameters [19]. MLR involved 25 animals and was repeated for each age level (T0–T22). Moreover, a residual analysis was performed to test if each regression model completely explained the studied association. Finally, the models were validated on a group of 13 animals, estimating their weight based on the body measurements provided by the respective model. The estimated weights were related to the real ones using a regression.

The Variance Infaction Factor (VIF) was evaluated to detect the presence of multicollinearity problems in the regression.

## 3. Results

A total of 45 animals were enrolled in the study, but seven were found to be sick at a measuring time (2 at 21 days, 1 at 49 days, 3 at 6 months old and 1 at 12 months old). Thus, a total of 38 rearing heifers were included: 25 animals represent the study group (SG), while 13 animals represented the validation group (VG). Descriptive statistics are reported in Table 1.

Prediction equations for BW and the linear effects of selected BMs at different times are reported in Table 2.

Results from the validation analysis are reported in Table 3.

Correlation analysis results between the different measures used for BW estimation are reported in Table 4.

The confidence interval regarding the relationship between real body weight and estimated body weight for each time investigated was reported in Figure 1 and Figure 2.

## 4. Discussion

Body weight is closely related to BM, with HG representing the most satisfactory single predictor of BW in cattle [13,20,21,22]. This method is cheap and accurate; thus, the literature has developed several predictive regression equations based on this parameter, alone or in combination with others [10,23,24]. Most of the research is old, focused on adult animals or was performed under less intensive production systems [9,10,24]; therefore, the aim of this observational study was to develop tools that could provide precise and meaningful descriptions of rearing dairy heifers’ phases of growth.

Monitoring heifer growth offers the most useful information to evaluate heifer performance throughout the rearing process, from birth to calving. Target weights and growth rates for animals of different ages are reported in the literature [3,25,26], suggesting that the optimal weight for Holstein heifers at first service can range between 341 and 400 kg [3,27]. Our results concerning BW at different evaluation times were in line with the literature [4,26,28,29], while data on WH were slightly fewer compared to previous studies [26]. Wilson and colleagues [10] evaluated BW and the same BM as assessed in our study in 826 Holstein bull calves; our results were slightly lower compared to the previously mentioned research [10]. This difference was probably related to the gender of the animals included (bull calves vs. heifers). The data of the present study represented a new evaluation of BW in Holstein heifers, which was especially important at 30 (T4), 180 (T13) and 450 (T22) days of life. Brickell and colleagues [30] reported that heifers showing a higher BW at these growing steps (30, 180 and 450 days old) reached puberty earlier and were bred earlier compared to the others. Authors concluded that BW evaluation at 30, 180 and 450 days old is directly related to age at first breeding and age at first calving, and of particular interest for farmers [30]. Considering this, it is easy to understand that monitoring heifer’s growth rate is crucial for early action, because poorly grown heifers usually require more services per conception, calve later and are more likely to be culled early [31].

Our results showed a high correlation between the analyzed BMs. However, the strong correlation is only present when all the BMs are evaluated, while when the BM are assessed individually, the correlation decreases considerably.

Equations estimating BW from BM were present in the literature [10,22,24]; however, our study showed equations specific to several different points of heifers’ growth. The developed equations were composed of different BMs at each evaluation time, highlighting the importance of having a specific, age-related formula. The regression analysis indicated that the use of HG was promising in all the equations built for BW estimation, followed by BL, WH and BCS, while SC and HW were included in only a few models. Heart girth and BCS are closely related to BW in both cows and growing calves [23,24,32], while the usefulness of BL and WH in BW estimation is controversial [5,10,24]. Differences between studies might depend on statistical analysis, as some recent papers related BL and WH alone to BW [5], while other research included several BMs in the model [10,24,33].

A limit of the present study is that all the included heifers underwent the same diet management, formulated with CNCPS, with 0.800 kg as an average weight gain target [16]; this diet permitted an ideal BW to be obtained for the first artificial insemination at 15 months. Although our growth rate target represented the average goal for Holstein rearing heifers, specific herd factors relating to nutritional management, body condition, or conformation genetics may limit the use of our equation for a herd with similar characteristics.

The main advantage in using equations from the present study is the estimation of BW without the need for a scale. However, special attention must be paid when assessing BCS and BM, because BCS is a subjective measure, which is superior as an indicator of fat reserves and the nutritional status of dairy animals [34]. Body measurements can vary due to the positioning and tension of the tape on the animal’s body. These problems can easily be overcome with some training and practice, which is easy to offer to most smallholder farmers.

The key times for BW assessment of heifers are at birth, at after weaning (around 60 days of life) and prior to the start of the service period (360–400 days). Birth weight allows future growth rates to be calculated accurately, while weaning weight is important for future performance evaluation. Birth weight mostly depends on the breed, and is usually similar throughout different farms, while weight at weaning varies between herds, because it is influenced by management [4]. Knowing the weight of animals prior to service is important, to assess if heifers enter the herd at the correct size and weight. Farmers would benefit from using few parameters for BW estimation during these key moments, for a timelier management. The results from the present study showed that equations for BW estimation at birth (T0) and at weaning time (T9) need only two parameters, and equations used at the beginning of the reproductive phase (T20-T22) need between 3 and 4 parameters. Thus, they can be considered feasible for monitoring heifers’ growth in a dairy farm.

The VG analysis led the farmers and the practitioners to be able to calculate the exact BW, instead of the estimated BW, starting from the BM.

The number of animals included in the present study is not high. Despite the use of VG analysis, which increases the feasibility of our results, the small group of heifers may represent a limitation. This aspect can create possible collinearity problems. For this reason, we calculated the VIF, which showed values between 1.005 and 3.207. In general, a VIF greater than 4 or a tolerance less than 0.25 indicates that multicollinearity may exist, and further investigation is required. Therefore, our data did not have multicollinearity problems. Increasing the sample size might lead to the establishment of more precise equations and could be especially useful in cases of high genetic variability inside a herd.

## 5. Conclusions

Body measurements of rearing heifers are highly related to their live BW. Our study showed some linear regression model equations, which can be used to estimate the rearing heifers’ weight at different ages, with 0.800 kg as an average weight gain target. These equations could be used under field conditions as a simple and cheap method with no need for a scale. The nutrition management used in the present study helped to obtain an ideal BW for a 15-month-old, at the point of first artificial insemination. Different growing rate targets may be studied in order to evaluate possible modifications to our equations.

## Figures and Tables

**Figure 1 animals-11-01846-f001:**
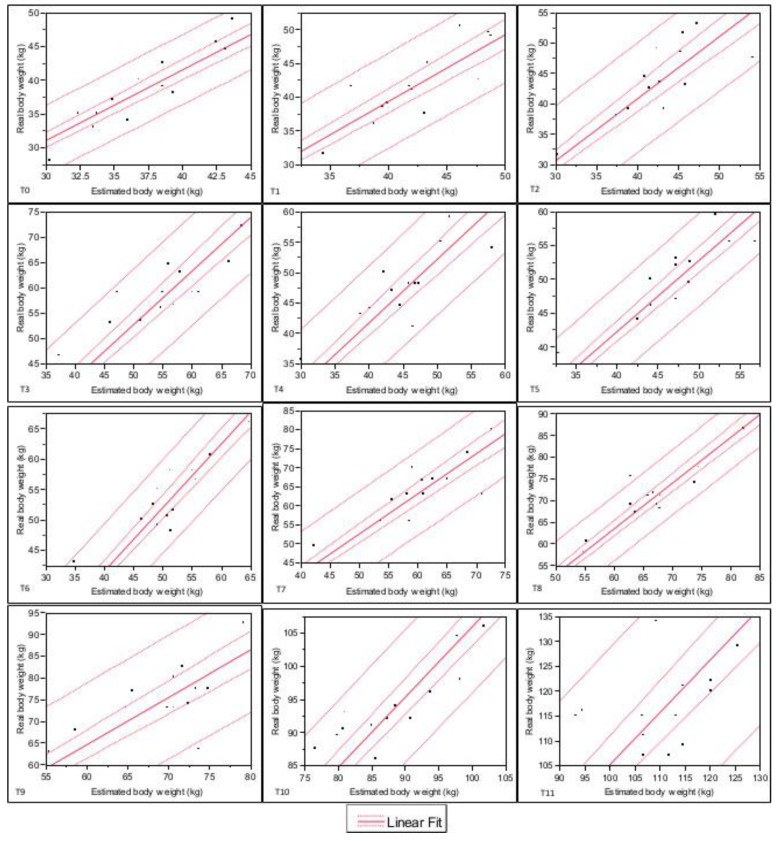
Confidence interval regarding ship between real body weight and estimated body weight investigated from T0 to T11. Internal dashed lines = Prediction lines. External dashed lines = Confidence lines. Legend: T0: at birth, T1: 7 days old, T2: 14 days old, T3: 21 days old, T4: 28 days old, T5: 35 days old, T6: 42 days old, T7: 49 days old, T8: 56 days old, T9: 2 months old, T10: 3 months old, T11: 4 months old.

**Figure 2 animals-11-01846-f002:**
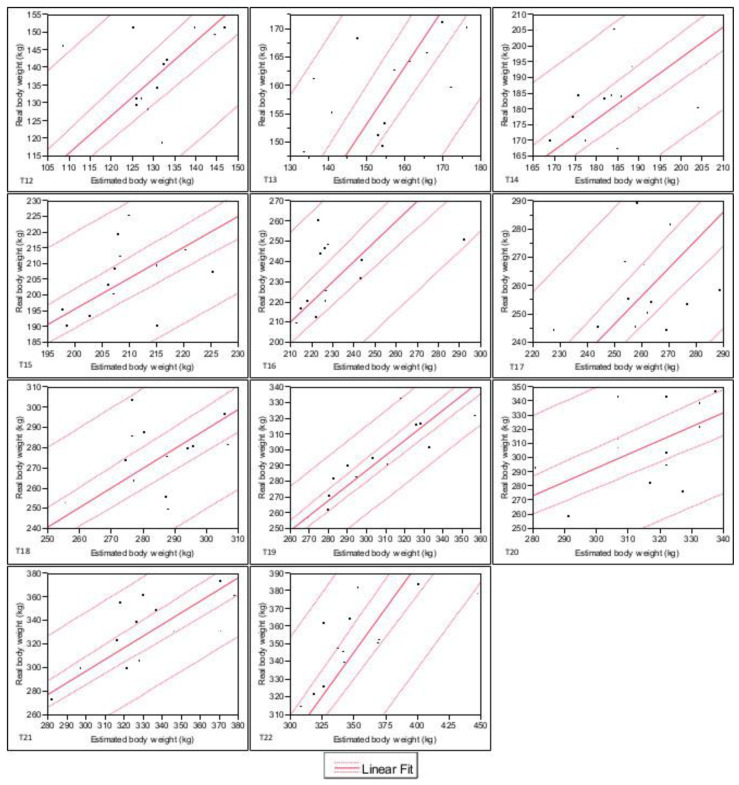
Confidence interval regarding relationship between real body weight and estimated body weight investigated from T12 to T22. Internal dashed lines = Prediction lines. External dashed lines = Confidence lines. Legend: T12: 5 months old, T13: 6 months old, T14: 7 months old, T15: 8 months old, T16: 9 months old, T17: 10 months old, T18: 11 months old, T19: 12 months old, T20: 13 months old, T21: 14 months old, T22: 15 months old.

**Table 1 animals-11-01846-t001:** Mean ± standard deviation of body weight, body measurements and body condition score evaluated in 25 rearing Holstein heifers at different times (T0–T22).

Time	BW (Kg)	WH (cm)	SC (cm)	HG (cm)	BL (cm)	HW (cm)	BCS
T0	39.50 ± 4.32	77.62 ± 2.77	12.04 ± 1.16	77.10 ± 3.49	69.04 ± 3.49	19.00 ± 1.76	2.68 ± 0.21
T1	43.38 ± 3.99	79.54 ± 2.71	12.16 ± 0.90	80.24 ± 3.09	71.12 ± 3.09	19.66 ± 1.36	2.79 ± 0.17
T2	46.42 ± 4.58	80.78 ± 2.43	12.64 ± 0.91	82.06 ± 3.77	73.02 ± 2.99	20.48 ± 1.57	2.85 ± 0.19
T3	49.34 ± 6.04	82.06 ± 2.61	12.78 ± 0.87	84.04 ± 3.97	74.98 ± 2.81	21.36 ± 1.50	2.83 ± 0.20
T4	52.22 ± 6.96	83.02 ± 3.18	12.88 ± 0.78	85.64 ± 3.60	76.18 ± 2.61	21.90 ± 1.55	2.86 ± 0.21
T5	55.56 ± 6.00	84.26 ± 3.29	13.12 ± 0.65	87.58 ± 3.11	78.66 ± 3.12	21.96 ± 1.11	2.92 ± 0.14
T6	59.60 ± 7.26	86.02 ± 3.46	13.04 ± 0.72	89.66 ± 4.52	79.84 ± 2.82	22.52 ± 1.18	2.90 ± 0.18
T7	64.14 ± 8.12	87.36 ± 3.35	13.46 ± 0.80	91.18 ± 4.00	81.20 ± 3.57	23.20 ± 2.26	2.97 ± 0.15
T8	69.64 ± 8.96	88.94 ± 3.98	13.52 ± 0.96	93.44 ± 4.51	83.08 ± 2.86	23.50 ± 2.27	2.98 ± 0.18
T9	74.92 ± 10.27	90.14 ± 4.11	13.86 ± 0.92	96.62 ± 5.33	84.26 ± 3.44	23.42 ± 1.56	2.97 ± 0.17
T10	102.70 ± 17.41	97.10 ± 6.42	14.02 ± 1.08	104.82 ± 6.68	91.96 ± 5.66	23.92 ± 1.58	3.06 ± 0.18
T11	128.56 ± 21.69	101.92 ± 6.10	14.68 ± 0.98	112.40 ± 7.54	98.88 ± 6.63	24.16 ± 2.54	3.12 ± 0.18
T12	149.92 ± 25.51	106.06 ± 5.94	15.16 ± 1.19	119.10 ± 7.45	105.12 ± 8.06	25.36 ± 3.29	3.14 ± 0.21
T13	179.50 ± 30.66	111.64 ± 7.07	16.06 ± 1.85	126.76 ± 8.11	110.94 ± 7.50	26.30 ± 3.48	3.22 ± 0.24
T14	205.78 ± 33.29	116.36 ± 7.12	16.24 ± 1.21	134.04 ± 10.17	116.18 ± 8.45	27.52 ± 3.93	3.24 ± 0.18
T15	231.98 ± 37.03	119.70 ± 6.04	16.98 ± 1.45	138.66 ± 9.61	121.02 ± 7.40	28.56 ± 4.51	3.18 ± 0.18
T16	252.88 ± 34.07	122.64 ± 6.78	17.94 ± 1.47	144.64 ± 9.37	124.44 ± 5.98	30.12 ± 3.23	3.14 ± 0.18
T17	272.94 ± 34.12	126.24 ± 6.43	18.90 ± 1.35	149.54 ± 9.03	127.08 ± 5.66	30.24 ± 3.31	3.18 ± 0.15
T18	299.42 ± 41.89	128.74 ± 6.68	19.94 ± 1.27	154.68 ± 9.21	130.38 ± 7.00	30.36 ± 2.77	3.10 ± 0.20
T19	321.58 ± 43.87	131.32 ± 5.74	20.94 ± 1.22	163.60 ± 22.11	134.80 ± 6.75	30.48 ± 2.09	3.15 ± 0.19
T20	350.04 ± 47.58	132.64 ± 5.35	21.82 ± 1.24	164.36 ± 8.50	138.48 ± 5.44	30.72 ± 2.40	3.21 ± 0.21
T21	378.00 ± 44.49	135.52 ± 4.10	22.70 ± 1.38	167.94 ± 7.41	142.64 ± 6.14	31.38 ± 2.15	3.20 ± 0.25
T22	404.00 ± 46.70	138.56 ± 4.54	23.64 ± 1.61	172.72 ± 7.88	145.96 ± 5.41	32.42 ± 2.20	3.21 ± 0.26

Legend: T0: at birth, T1: 7 days old, T2: 14 days old, T3: 21 days old, T4: 28 days old, T5: 35 days old, T6: 42 days old, T7: 49 days old, T8: 56 days old, T9: 2 months old, T10: 3 months old, T11: 4 months old, T12: 5 months old, T13: 6 months old, T14: 7 months old, T15: 8 months old, T16: 9 months old, T17: 10 months old, T18: 11 months old, T19: 12 months old, T20: 13 months old, T21: 14 months old, T22: 15 months old. WH: wither height, SC: shin circumference, HG: heart girth circumference, BL: body length, HW: hip width, BCS: body condition score.

**Table 2 animals-11-01846-t002:** Prediction equations of BW and linear effects of selected BMs evaluated in 25 rearing Holstein heifers at different times (T0–T22).

Time	Prediction Equations	Constant	WH	SC	HG	BL	HW	BCS	*p*-Value	Ac. R^2^%
T0	Y = a + b_3_X_3_ + b_6_X_6_	−56.747 ***			1.014 ***			6.782 ***	***	0.836
T1	Y = a + b_1_X_1_ + b_3_X_3_ + b_4_X_4_ + b_6_X_6_	−67.402 ***	0.378		0.517 **	0.389 **		4.121	***	0.688
T2	Y = a + b_1_X_1_ + b_3_X_3_ + b_4_X_4_ + b_5_X_5_	−68.033 ***	0.569 *		0.674 **	0.344	−0.589		***	0.707
T3	Y = a + b_3_X_3_ + b_4_X_4_ + b_5_X_5_ + b_6_X_6_	−73.605 ***			1.030 ***	0.524 **	−0.888 *	5.840 *	***	0.784
T4	Y = a + b_2_X_2_ + b_3_X_3_ + b_5_X_5_ + b_6_X_6_	−76.642 ***		−1.302	1.557 ***		−0.587	8.961 **	***	0.770
T5	Y = a + b_1_X_1_ + b_3_X_3_ + b_4_X_4_ + b_6_X_6_	−100.266 ***	0.499 *		0.602 **	0.445 **		9.063 *	***	0.777
T6	Y = a + b_3_X_3_ + b_4_X_4_ + b_6_X_6_	−104.225 ***			0.863 ***	0.794 ***		8.122 *	***	0.756
T7	Y = a + b_1_X_1_ + b_2_X_2_ + b_3_X_3_ + b_4_X_4_	−125.697 ***	0.510	1.271	1.042 ***	0.422			***	0.726
T8	Y = a + b_1_X_1_ + b_2_X_2_ + b_3_X_3_ + b_4_X_4_	−132.888 ***	0.472	1.406	1.173 ***	0.397			***	0.772
T9	Y = a + b_1_X_1_ + b_3_X_3_	−113.875 ***	0.902 ***		1.124 ***				***	0.787
T10	Y = a + b_1_X_1_ + b_2_X_2_ + b_3_X_3_ + b_4_X_4_ + b_6_X_6_	−198.994 ***	0.834 ***	1.546	1.226 ***	0.448		9.803	***	0.888
T11	Y = a + b_1_X_1_ + b_3_X_3_ + b_4_X_4_ + b_5_X_5_	−175.638 ***	0.651		1.308 **	0.556	1.5242		***	0.854
T12	Y = a + b_1_X_1_ + b_3_X_3_ + b_4_X_4_	−205.406 ***	0.805		1.320 *	1.088			***	0.790
T13	Y = a + b_1_X_1_ + b_3_X_3_ + b_4_X_4_	−263.825 ***	1.230 **		1.508 **	1.044 *			***	0.846
T14	Y = a + b_3_X_3_ + b_4_X_4_ + b_6_X_6_	−299.391 ***			2.051 ***	1.470 ***		18.251 *	***	0.894
T15	Y = a + b_3_X_3_ + b_4_X_4_ + b_5_X_5_	−267.306 ***			2.226 ***	1.099 *	1.975 *		***	0.853
T16	Y = a + b_1_X_1_ + b_3_X_3_	−253.243 ***	1.681 **		2.068 ***				***	0.754
T17	Y = a + b_1_X_1_ + b_2_X_2_ + b_3_X_3_ + b_4_X_4_ + b_6_X_6_	−516.279 ***	1.299	5.118 **	1.445 *	1.923 **		21.331	***	0.782
T18	Y = a + b_3_X_3_ + b_4_X_4_	−387.019 ***			3.293 ***	1.347 *			***	0.776
T19	Y = a + b_3_X_3_ + b_4_X_4_ + b_6_X_6_	−506.335 ***			0.606 **	3.591 ***		76.624 ***	***	0.676
T20	Y = a + b_3_X_3_ + b_4_X_4_ + b_6_X_6_	−677.207 ***			4.008 ***	1.926 *		31.457 *	***	0.839
T21	Y = a + b_3_X_3_ + b_4_X_4_ + b_5_X_5_ + b_6_X_6_	−534.638 ***			3.272 ***	2.691 ***	−3.859 **	31.328 *	***	0.853
T22	Y = a + b_1_X_1_ + b_3_X_3_ + b_4_X_4_ + b_5_X_5_	−649.723 ***	1.916		3.367 ***	2.112 *	−3.192 **		***	0.871

Legend: T0: at birth, T1: 7 days old, T2: 14 days old, T3: 21 days old, T4: 28 days old, T5: 35 days old, T6: 42 days old, T7: 49 days old, T8: 56 days old, T9: 2 months old, T10: 3 months old, T11: 4 months old, T12: 5 months old, T13: 6 months old, T14: 7 months old, T15: 8 months old, T16: 9 months old, T17: 10 months old, T18: 11 months old, T19: 12 months old, T20: 13 months old, T21: 14 months old, T22: 15 months old. WH: wither height, SC: shin circumference, HG: heart girth circumference, BL: body length, HW: hip width, BCS: body condition score. * = *p*-value between 0.05 and 0.01; ** = *p*-value between 0.01 and 0.001; *** = *p*-value < 0.001.

**Table 3 animals-11-01846-t003:** Linear regression evaluating the relation between the estimated body weight and the real body weight using the validation group (*n* = 13 rearing heifers).

Time	Prediction Equations	*p*-Value	R^2^%	RMSE
T0	Y = 1.038X	***	0.99	2.28
T1	Y = 0.986X	***	0.99	3.13
T2	Y = 1.024X	***	0.99	4.02
T3	Y = 1.046X	***	0.99	4.11
T4	Y = 1.057X	***	0.99	3.02
T5	Y = 1.042X	***	0.99	3.42
T6	Y = 1.057X	***	0.99	4.81
T7	Y = 1.056X	***	0.99	4.95
T8	Y = 1.059X	***	0.99	3.33
T9	Y = 1.079X	***	0.99	6.27
T10	Y = 1.061X	***	0.99	4.42
T11	Y = 1.053X	***	0.99	10.38
T12	Y = 1.054X	***	0.99	12.69
T13	Y = 1.021X	***	0.99	11.37
T14	Y = 0.982X	***	0.99	11.60
T15	Y = 0.978X	***	0.99	10.73
T16	Y = 1.000X	***	0.99	19.61
T17	Y = 0.986X	***	0.99	18.11
T18	Y = 0.964X	***	0.99	17.39
T19	Y = 0.956X	***	0.99	12.47
T20	Y = 0.976X	***	0.99	25.15
T21	Y = 0.990X	***	0.99	22.04
T22	Y = 0.987X	***	0.99	25.77

Legend: Y = real body weight, X = estimated body weight, T0: at birth, T1: 7 days old, T2: 14 days old, T3: 21 days old, T4: 28 days old, T5: 35 days old, T6: 42 days old, T7: 49 days old, T8: 56 days old, T9: 2 months old, T10: 3 months old, T11: 4 months old, T12: 5 months old, T13: 6 months old, T14: 7 months old, T15: 8 months old, T16: 9 months old, T17: 10 months old, T18: 11 months old, T19: 12 months old, T20: 13 months old, T21: 14 months old, T22: 15 months old. *** = *p*-value < 0.001, RMSE: Root mean square error.

**Table 4 animals-11-01846-t004:** Values concerning the coefficient of correlation between different measures used for BW estimation in 25 rearing Holstein heifers.

	BW	WH	SC	HG	BL	HW	BCS
BW	1.00	0.97	0.93	0.98	0.97	0.86	0.60
WH	−0.05	1.00	0.90	0.99	0.98	0.88	0.64
SC	0.32	−0.04	1.00	0.92	0.91	0.80	0.53
HG	0.60	0.42	0.02	1.00	0.99	0.89	0.63
BL	0.12	0.43	0.12	0.28	1.00	0.89	0.63
HW	−0.19	0.03	−0.03	0.23	0.15	1.00	0.59
BCS	−0.08	0.13	−0.14	0.11	−0.02	−0.01	1.00

Above the diagonal are the results obtained with Pearson correlation, while below the diagonal are the results obtained with partial correlation. Legend: BW: body weight, WH: wither height, SC: shin circumference, HG: heart girth circumference, BL: body length, HW: hip width, BCS: body condition score.

## Data Availability

The data presented in this study are available in article.

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
