# Peer review of "Designing Statistical Models for Holstein Rearing Heifers’ Weight Estimation from Birth to 15 Months Old Using Body Measurements"

_animals, 2021, doi:10.3390/ani11071846_

Round 1

Reviewer 1 Report

I accept the minor revisions of the manuscript.

Author Response

Thank you

Reviewer 2 Report

I have recently revised the manuscript animal 1256234 it again and my response to the comments are satisfactory

Author Response

Thank you

Reviewer 3 Report

I understand that with the changes made by the authors, the work has improved compared to the first version.

However, I believe that the statistical analysis should address possible multicollinearity problems, making use of the Variance Infaction Factor (VIF) measures.

I think that among the limitations of the work, the small size of the analyzed sample should be commented.

Author Response

I believe that the statistical analysis should address possible multicollinearity problems, making use of the Variance Infaction Factor (VIF) measures.

AU: The main text has been modified (see lines 289-292).

The small size of the analyzed sample should be commented.

AU: The main text has been modified (see lines 287-294).

This manuscript is a resubmission of an earlier submission. The following is a list of the peer review reports and author responses from that submission.

Round 1

Reviewer 1 Report

Well written paper on an intersting subject. I would prefer some more explanation on the statistacal modelling, but all in all I found the text easy to follow and understand.

Author Response

Line 199 abbreviation AM is not explained in the text

AU: AM is the abbreviation of Alessio Madrigali, one of the co-authors. He performed the body measurements.

Line 122 abbreviation LT is not explained in the text

AU: LT is the abbreviation of Luca Turini, one of the authors. He performed the body measurements.

Line 134-135 please give a brief explanation or references to how the model selection was performed.

AU: We provided a more detailed explanation of statistical analysis (see lines 141-162).

Line 139 Unclear last sentence please rewrite, specify that this is a validation group (VG) and that it is heifers.

AU: Done (see lines 165-168).

I miss an explanation of how the validation analysis was performed.

AU: We provided a more detailed explanation of statistical analysis (see lines 165-168).

Line 200 “performed in developing countries”, I would prefer using “research performed under less intensive production systems”.

AU: Done (see line 228).

Reviewer 2 Report

The body measurement information for evaluating body weight and growing rate in dairy heifers is very comprehensive, but the objetives are not achieves due to the lack of and updates methodology, sucha as that based on nonlinear models such as logistic models, Gomperez or Richard´s

to meet the objetives set, they should have used a methodology to evaluate te model efficiency, such as ROC (Receiver Operative Characteristics).

The body weight and other measures recorded during 22 times were analyzed as independent, using linear regression for each time, however with non linear model they would be able to integrate all informatión and their contribution would been greater.

Author Response

The body measurement information for evaluating body weight and growing rate in dairy heifers is very comprehensive, but the objectives are not achieves due to the lack of and updates methodology, such as that based on nonlinear models such as logistic models, Gomperez or Richard´s to meet the objectives set, they should have used a methodology to evaluate the model efficiency, such as ROC (Receiver Operative Characteristics).

AU: Our models are not based on a non-linear regression, precisely because we want to see if it is possible to estimate the weight of the animals on the basis of other more easily determined parameters such as body measurements. Therefore, it is not necessary to determine the ROC. In our work each regression model was evaluated by statistical test, R-squared calculation and residual analysis. A more detailed description was provided in the materials and methods.

The body weight and other measures recorded during 22 times were analyzed as independent, using linear regression for each time, however with non-linear model they would be able to integrate all information and their contribution would been greater.

AU: The aim of the work is to estimate the weight of the animal by evaluating other parameters. Therefore, linear regression is crucial.

Reviewer 3 Report

The aim of this study was to estimate rearing heifers weight based on their body dimension characteristics.

In my opinion, the subject studied in this work is of great interest from the point of view of applied research as well as from the field of dairy farm management. The work is well written.

However, I believe that the authors should more precisely express the motivation, purpose and contribution of the work in the Introduction. As well, the Discussion of results and Conclusions must be improved to make a clearer and more interesting contribution to the literature.

The Introduction should justify in more depth the interest of the work, showing greater attention to the previous literature on the subject and explaining in more detail and clarity the contribution of the paper to previous knowledge and research.

Regarding the Methodology, the validity of the regressions carried out with a reduced number of cases must be justified, and possible multicollinearity problems must also be addressed. In this sense, high correlations are observed between some explanatory variables such as HG and WH, HG and SC (Table 4).

The discussion and conclusions sections should be improved. For example, the results obtained refer to the analysis of animals with a specific treatment (feeding,...). The applicability of these results in different handling and feeding conditions should be discussed.

How valid would the application of these results be to a specific farm?

Should this analysis be performed on each farm for the results to be valid for predicting the weight of heifers on that farm?

The conclusions should also comment on the limitations of the work and possible future lines of work to overcome these limitations.

Author Response

The Introduction should justify in more depth the interest of the work, showing greater attention to the previous literature on the subject and explaining in more detail and clarity the contribution of the paper to previous knowledge and research.

AU: The main text has been modified (see lines 68-73).

Regarding the Methodology, the validity of the regressions carried out with a reduced number of cases must be justified, and possible multicollinearity problems must also be addressed. In this sense, high correlations are observed between some explanatory variables such as HG and WH, HG and SC (Table 4).

AU: We agree with the reviewer. However, as reported in Table 4, the partial correlations between variables are much lower than the Pearson correlations. This is sufficient to reduce multicollinearity problems.

The discussion and conclusions sections should be improved. For example, the results obtained refer to the analysis of animals with a specific treatment (feeding,...). The applicability of these results in different handling and feeding conditions should be discussed.

How valid would the application of these results be to a specific farm?

Should this analysis be performed on each farm for the results to be valid for predicting the weight of heifers on that farm?

AU: The main text has been modified (see lines 262-268).

The conclusions should also comment on the limitations of the work and possible future lines of work to overcome these limitations.

AU: The main text had been modified (see lines 296-300).
